# Targeting the Glutaminolysis Pathway in Glaucoma-Associated Fibrosis

**DOI:** 10.3390/ijms27010012

**Published:** 2025-12-19

**Authors:** Áine Kelly, Mustapha Irnaten, Colm O’Brien

**Affiliations:** 1Department of Ophthalmology, Mater Misericordiae University Hospital, D07 R2WY Dublin, Ireland; 2Catherine McAuley Research Centre, University College Dublin, D07 KX5K Dublin, Ireland

**Keywords:** fibrosis, glutaminolysis, glaucoma, bioenergetics

## Abstract

Glaucoma is a group of progressive optic neuropathies and the leading cause of irreversible vision loss worldwide. It is a chronic eye disease, and its major pathological features include fibrosis of the trabecular meshwork, Schlemm’s Canal and lamina cribrosa. Central to fibrosis is extracellular matrix (ECM) remodelling and metabolic reprogramming. Glutaminolysis is an alternative energy pathway that has previously been shown to be implicated in the metabolic reprogramming associated with cancer and other fibrotic diseases, facilitating ECM remodelling and cell proliferation. This paper reviews fibrosis, glutaminolysis in the setting of fibrosis, and fibrosis and glutaminolysis in the context of glaucoma. We review the evidence for fibrosis and metabolic reprogramming in oncology and systemic fibrotic diseases, which reveals a predilection for glutaminolysis. We review the current therapies that exist to target these pathways, and find glutaminolysis to be a potential target for future therapies in glaucoma.

## 1. Introduction

### 1.1. Glaucoma

Glaucoma is a group of progressive optic neuropathies and the leading cause of irreversible vision loss worldwide [1]. It is a chronic eye disease, and its major pathological features include fibrosis of the trabecular meshwork (TM) and lamina cribrosa (LC) and neurodegeneration of the retinal ganglion cell layer and axons of the optic nerve head [2]. With age being a major risk factor, its incidence is forecast to increase to 111.8 million by 2040, highlighting the significance of its growing disease burden on the ageing global population [3]. In addition, glaucoma generally has an insidious onset, where patients experience vague and non-specific symptoms, if at all, which leads to a delay in presentation [4]. This poses a challenge to clinicians as the current pillars of treatment are based on preventing further damage and visual loss as opposed to recovering a previous visual standard [5]. This underscores the importance of early detection and timely management in glaucoma. Increased intraocular pressure (IOP), a significant risk factor, can cause mechanical distortion in the connective tissue of the LC of the optic nerve head (ONH) [6], while the resulting increased extracellular matrix (ECM) deposition leads to tissue fibrosis [7]. These fibrotic changes within the LC and ONH may directly influence axonal damage [8].

### 1.2. Fibrosis

Fibrosis is a dysregulated and chronic state of wound healing that causes the overgrowth, hardening or scarring of tissues [9]. This is attributed to an excess deposition of ECM components, tissue remodelling and increased tissue stiffness [10]. This progressive remodelling is the end result of a chronic inflammatory reaction that can be triggered by a variety of stimuli, including chemical insults, mechanical tissue injury and hypoxia [11], and ultimately can lead to permanent scarring, organ malfunction and death [12]. Although fibrosis may be initiated by a range of triggers, the common step is the activation of ECM-producing myofibroblasts, which contribute significantly to the remodelling of fibrotic tissue [13]. This remodelling, and the subsequent parenchymal damage, is often associated with an inflammatory response, and thus leads to the secretion of various cytokines by immune cells.

### 1.3. Various Cell Types Associated with Fibrosis

Multiple cell types are integral to the propagation of fibrosis, and almost all immune cell types are involved in some way [14]. Along with myofibroblasts, mentioned above, macrophages and their haematological precursor, monocytes, play an important regulatory role in fibrosis [15]. The diversity and plasticity associated with macrophages [16] are at odds with the stiff, over-proliferation associated with fibrosis. However, their production of an inflammatory response, with the release of inflammatory chemokines such as Tumour Necrosing Factor-α, (TNF-α) and interleukin-1, -6 or -12, is an important host reaction to a hostile stimulus. When this reaction is not resolved, it can lead to chronic inflammation and fibrosis. Alternative types of macrophages can contribute to pathological scarring, propagating fibrosis [17]. In addition, it has recently been found that uncontrolled pro-inflammatory macrophages can directly contribute to the worsening of fibrosis through macrophage–myofibroblast transition (MMT) [18]. The role of T cells in fibrosis is diverse, and different subgroups contribute to both the development and resolution of fibrosis. Their role is often tissue- and disease-specific. Th1 cells produce pro-inflammatory cytokines that may attenuate fibrosis, but may play either a profibrotic [19] or antifibrotic role, such as that seen in cardiac fibrosis [20]. Th2 cells can directly support fibrosis by stimulating fibroblasts in order to allow production of collagen with IL-4 and IL-13 [21]. Th1 and Th2 T cells have been associated with scar-associated fibrosis [20]. Other types, such as regulatory T cells, may limit fibrosis by promoting tissue repair [22].

### 1.4. Mechanism of Fibrosis

As outlined above, fibrosis is the end-stage result of chronic inflammation that is often associated with a loss of function of that particular tissue. When normal tissue repair becomes excessive, the equilibrium between ECM production and degradation is pathologically altered [23]. In healthy tissue, the ECM functions as a dynamic network of tissue-specific macromolecules that provides structural support for cells and tissues, as well as regulates cell signalling and functions [24]. In fibrosis, activated fibroblasts (myofibroblasts) express α-smooth muscle actin (α-SMA), a protein usually associated with smooth muscle cells. They increase the expression of fibrillar collagen (Type I, III, V, VI), and support other ECM macromolecules. In addition, they inhibit ECM-degrading enzymes [25]. This leads to an increased, uncontrolled secretion of ECM, a stiffer cell microenvironment and, ultimately, loss of tissue function.

### 1.5. Similarities Between Fibrosis and Cancer

Fibrosis and cancer share multiple similarities, not only in their accumulation of and changes within the ECM, but also in the presence of genetic alterations and metabolic reprogramming [26]. Fibroblast proliferation and sustained differentiation are pathophysiological mechanisms common to both fibrosis and cancer. This in turn leads to ECM modifications and increased ECM stiffness [27,28]. With regard to genetic alterations, both fibrosis and cancer have been found to have increased transforming growth factor-beta (TGF-β) production and cellular senescence [29]. In addition, epithelial–mesenchymal transition (EMT) is a biological process where epithelial cells transform both phenotypically and functionally to mesenchymal-like cells [30], and this is a process common to both entities. Profibrotic and pro-inflammatory cytokines support the process of EMT in fibrosis [31]. Likewise, in cancer, EMT supports the proliferation, seeding and development of metastasis [32]. The presence of these shared properties is supported by the pharmaceutical exploitation of such, such as the targeting of the AXL tyrosine kinase receptor. This has been shown to reduce fibrosis [33], while also exhibiting promising antitumour effects [34]. Metabolic reprogramming is intrinsic to the proliferation of cancer cells, and is also common to fibrosis. This will be discussed in detail below.

## 2. Metabolic Reprogramming in Fibrosis and Cancer

### 2.1. Metabolic Reprogramming in Fibrosis

Profibrotic cells can exhibit metabolic reprogramming through alternative pathways under certain circumstances. These cells exhibit an increased proliferation rate and mitochondrial dysfunction [35]. These alterations include a decreased rate of oxidative phosphorylation and an increased reliance on glutaminolysis, glycolysis and one-carbon metabolism in order to match the increased metabolic demands through alternative pathways [36]. Of note, one-carbon metabolism plays a significant role in providing one-carbon molecules for nucleotide synthesis and reductive metabolism [37]. It involves multiple energy-producing cycles, including the folate and methionine cycles, and produces one-carbon molecules to act as precursors for methylation reactions. The recent literature has suggested that a similar mechanism of increased reliance on alternative energy pathways occurs in the fibroblast during activation, an important component of glaucoma-associated fibrosis [38]. Glutaminolysis is a metabolic pathway that involves glutamine uptake through transmembrane proteins to maintain cellular bioenergetics and replenish macromolecular biosynthesis, with its products supplying the Citric Acid Cycle [39]. It has been frequently reported in organ fibrosis [40], and contributes to the synthesis of the two most abundant amino acids in collagen, glycine and proline [41]. Currently, research on glutaminolysis is mainly limited to various types of cancer. This article aims to review glutaminolysis, its contribution to fibrosis and its role in glaucoma-associated fibrosis.

### 2.2. Mitochondrial Dysfunction

Central to the propagation of fibrosis is mitochondrial dysfunction [42]. In a healthy cell, the mitochondria, also known as ‘the power house’ of cells, maintain and control important basic functions including energy metabolism, signalling transduction, apoptosis and cell differentiation [43]. They also play a critical role in controlling the cellular balance of reactive oxygen species [ROS]. However, in fibrotic tissue, a damaging microenvironment of oxidative stress, inflammation and hypoxia results in the perfect storm for mitochondrial dysfunction to occur [42]. This leads to increased ROS and resulting oxidative damage to mitochondrial proteins and DNA [44], further supporting the uncontrolled proliferation of ECM.

### 2.3. Need for Metabolic Reprogramming Due to Increased Energy Demands

The increased bioenergetic and biosynthetic demands during fibrosis and fibroblast activation necessitate dramatic cell metabolic changes to meet these increased demands. Principal among these metabolic changes in fibrosis are an increased reliance on glycolysis and glutaminolysis as opposed to oxidative phosphorylation, which is used under normal conditions to provide enough ATP for regular cell function [36]. An increased reliance on aerobic glycolysis, or the ‘Warburg Effect’, has been extensively studied in the oncology literature, and we will discuss this further below. With regard to the increased glycolysis in fibrosis, a study by Xie and colleagues measured lactate as a surrogate for glycolytic rate in lung myofibroblasts and found glycolysis to be an essential step in the development of lung fibrosis. They demonstrated that it was required not only for the initiation, but also for the sustainment of myofibroblast differentiation [45]. These results were supported in renal myofibroblast activation, where enhanced aerobic glycolysis was found to provide sufficient ATP and also act as a source of metabolic intermediates. This switch from oxidative phosphorylation to aerobic glycolysis was hypothesised to be a necessary step in the proliferation of fibroblasts [46].

In addition to increased glycolysis, the metabolic reprogramming associated with fibrosis also reveals an increased rate of glutaminolysis. Glutamine, the most abundant amino acid found in plasma [47], is converted to glutamate by the rate-limiting enzyme glutaminase (GLS). Multiple studies have found GLS to be upregulated in fibrotic environments, including the lung in interstitial lung disease [48], the liver in nonalcoholic steatohepatitis [49] and the endometrium in endometriosis [50]. This is significant as not only is glutaminolysis an important producer of ATP in fibrotic environments, but it also generates both proline and glycine, which together contribute over 50% of the amino acids in collagen, and thus further contribute to the proliferation of the ECM [51].

### 2.4. Metabolic Reprogramming in Cancer

Since Otto Warburg first described the capacity of tumour cells to sustain their energy requirements through obtaining ATP through alternative sources other than oxidative phosphorylation [52], the interest in glutaminolysis as an alternative metabolic pathway has garnered much attention. As was elegantly described by Dvorak as a ‘wound that does not heal’ [53], the recognised significant fibrotic component to cancer is intrinsically intertwined with metabolic reprogramming and glutaminolysis. In addition, tumour cells can manipulate their utilisation of glutamine and use it for citrate production through reductive carboxylation of the TCA cycle [54].

Cancer cells have shown significant dependence on glutamine, and certain types have been shown to be unable to survive without it [55]. The use of glutamine as both a nitrogen donor and as a carbon source that feed into the TCA cycle via glutaminolysis generates key biosynthetic intermediates, and has been shown to be imperative to tumour cells. This has been termed ‘glutamine addiction’. As outlined above, the deamination of glutamine to glutamate mediated by GLS is the rate-limiting step in glutaminolysis. GLS1 has increased expression in various cancer cells. Huang et al. found GLS1 to be significantly upregulated in colorectal cancer, and its expression to be associated with later staging of the disease, distant metastasis, and deeper infiltration of tissue [56].

Alternative metabolic pathways that are utilised in cancer include glycolysis and one-carbon metabolism. The upregulation of glycolysis has been extensively researched in the setting of oncology. It produces adenosine triphosphate (ATP) by the breakdown of glucose to lactate under anaerobic conditions [57]. It serves as an alternative energy pathway favoured by cancer cells, even in oxygen-sufficient environments [58], and allows them to meet their increased metabolic demands, as referenced above.

One-carbon metabolism has been found to be upregulated in cancer cells [59]. There are multiple pathways that can generate one-carbon, or methyl units, as they are otherwise known [60]. One-carbon units can be utilised for a variety of functions in cancer, including nucleotide synthesis, methylation pathways, and the production of nicotinamide adenine dinucleotide (NADH) and nicotinamide adenine dinucleotide phosphate (NADPH) that serve as important sources of electrons in redox reactions [61]. These pathways, which are intertwined with folate and methionine cycles, are currently targeted by chemotherapy drugs, including methotrexate and 5-fluorouracil [62,63,64].

### 2.5. Metabolic Manipulation of Cell Death and Apoptosis

Under normal conditions, the completion of healing is associated with the apoptosis of myofibroblasts [65]. However, in pathological conditions, there is an inherent resistance of the myofibroblasts to apoptosis, previously dubbed the ‘hallmark of fibrotic disease’ [66]. This results in metabolic reprogramming and upregulated cytokine stimulation. Sinha, Sparks and colleagues recently investigated fibroblast inflammatory priming and its effect on apoptosis in different types of reindeer skin, namely regeneration-primed fibroblasts found in reindeer velvet (antler skin) and inflammatory-primed fibroblasts found in reindeer back skin. They investigated metabolic reprogramming, the importance of fibroblastic priming and its development into a profibrotic, scar-inducing fibroblast or a regenerative, pro-healing fibroblast. Their study revealed that the velvet type regenerated, whereas the back skin formed a fibrotic scar. Further genetic investigation found that back-skin fibroblasts expressed pro-inflammatory cytokines mimicking profibrotic adult human and rodent fibroblasts by amplifying myeloid infiltration during repair and exhibiting metabolic reprogramming, whereas velvet fibroblasts adopted an immunosuppressive phenotype [67]. Metabolic adaptations are also seen in profibrotic conditions, where the fibroblasts have a higher rate of ATP synthesis, and glycolysis is the primary energy source as opposed to oxidative phosphorylation [68].

## 3. Glutaminolysis

### 3.1. Glutamine

Glutamine, although considered an unessential amino acid in physiological conditions, is the most abundant free amino acid in plasma and plays a fundamental role in cell metabolism, as it acts as a ready source in a variety of biochemical processes and, more notably, as a source in nutrient-poor microenvironments [55]. It has integral roles, such as being a precursor for glutathione synthesis, proteins, nucleic acids and lipids, as well as serving as an alternative carbon source for the Tricarboxylic Acid (TCA) cycle, and so is a prominent component in ATP synthesis [69]. Glutamine is also integral to regulating redox homeostasis by supplying glutamate for the synthesis of glutathione (GSH), a major antioxidant. Glutathione has been extensively studied as a protective factor against oxidative stress, which works by removing reactive oxygen species [70]. The excitotoxic properties of glutamate have been extensively studied in the literature.

### 3.2. The Common Pathway in Glutaminolysis

Glutaminolysis is a metabolic pathway that involves glutamine uptake and catabolism to maintain bioenergetics and replenish macromolecular biosynthesis (Figure 1). The extracellular glutamine is transported into the cytoplasm by plasma membrane glutamine transporters, namely SLC1A5, SLC38A1 and SLC38A2. In order to reach the mitochondria, glutamine is then transported into the mitochondria by the SLC1A5 variant, a mitochondrial gene transporter [71]. Two isoenzymes of glutaminase, GLS1 and GLS2, that were initially discovered in the kidney and liver, respectively, convert glutamine to both glutamate and ammonia. This is the rate-limiting reaction of glutaminolysis [72]. Glutamate is subsequently converted to α-ketoglutarate (α-KG) by two different pathways: the first occurs through the activity of glutamate dehydrogenase (GDH) and the second through the activity of a group of aminotransferases [73]. It is then either used as a source in the TCA cycle to produce ATP or in macromolecular biosynthesis. In essence, glutamine travels through this pathway to the TCA as follows:


l-glutamine → l-glutamate ⇆ α-ketoglutarate → tricarboxylic acid (TCA) cycle


### 3.3. The Glutaminase II Pathway

There is an alternative, less studied pathway known as the glutaminase II pathway, more recently referred to by Cooper et al. as the glutamine transaminase-ω-amidase (GTωA) pathway [75]. Glutamine also travels through this cascade to the TCA cycle, but is first transaminated to α-ketoglutaramate (KGM) and further hydrolyzed by ω-amidase to α-ketoglutarate [76]. α-ketoglutarate (α-KG) plays an active role in both fatty acid biosynthesis and NADH production.

### 3.4. Nucleotide Synthesis Associated with Glutaminolysis

Cytosolic glutamine plays a central role in supporting nucleotide production [77]. The γ-nitrogen of glutamine is donated in five distinct reactions during de novo nucleotide synthesis, meaning that its availability directly regulates both purine and pyrimidine biosynthesis. In the purine pathway, two molecules of glutamine are required to form inosine monophosphate (IMP), the common precursor for adenosine monophosphate (AMP) and guanosine monophosphate (GMP) [78]. A further glutamine molecule is specifically needed to convert IMP into GMP. In pyrimidine synthesis, glutamine is first used by carbamoyl phosphate synthetase (CPS1 in mitochondria or CPS2 in the cytosol) to initiate the pathway [79]. Another glutamine molecule is then consumed in the conversion of uridine triphosphate (UTP) into cytidine triphosphate (CTP). The role of glutamine and glutaminolysis in nucleotide synthesis is extensive, and in addition to that mentioned above, previous studies have shown glutamine can also utilise alternate pathways to support nucleotide synthesis, such as with aspartate, which occurs when glutamate is formed from the transamination of glutamine [80].

### 3.5. Neurodegeneration

Glutamate, of which glutamine is the main precursor, is the primary excitatory neurotransmitter in the brain and has many roles in the function of the central nervous system [81]. It is inextricably linked to glutaminolysis, as glutamine is deaminated in mitochondria by glutaminase to produce glutamate [82]. Glutamatergic dysregulation has long been recognised as a significant factor in numerous disorders, including psychiatric, neurodevelopmental and neurodegenerative diseases [83,84,85]. An excess of glutamate can flood neuronal glutamate receptors and ultimately lead to neuronal oxidative stress [86].

Its toxicity and the resulting neurodegeneration have been known to exist within the eye since the work of Lucas and Newhouse in 1957, who discovered its toxic effect on the inner layers of the retina [87]. The exact mechanism of this so-called excitotoxicity is unclear. Previously, varying levels of glutamate were found to cause both apoptotic and necrotic injury [88]. Wu et al. recently revisited its role in excitotoxicity, and found it to be multi-factorial, involving calcium imbalance, increased oxidative stress, the resulting increase in misfolded proteins and the induction of apoptosis, as well as the disruption of cellular homeostasis [89].

Although its role has been previously investigated in the setting of glaucoma, no definitive results or therapeutic targets have been discovered. However, this paper will focus on the role of glutaminolysis in metabolic reprogramming, and not its relation to the excitotoxic properties of glutamate, for which there is extensive existing literature.

## 4. Glutaminolysis in Oncology

### 4.1. The Role of Glutaminase 1

The increased reliance of tumour cells on glutamine metabolism is linked to Hypoxia-Inducible Factor 1-alpha (HIF-1α) activity. Xiang et al. dived deeper into GLS1 in colorectal cancer, and found it to be influenced by these well-known adverse prognostic factors. They found GLS1 expression to be induced by hypoxia and to be dependent on HIF-1a. They found a significantly decreased survival rate as well as an increased correlation with lymph node metastasis and advanced clinical stage in tumours where GLS1 had an increased expression when compared to tumours with a lower GLS1 level [90]. In head and neck squamous cell carcinomas, Yang et al. supported the overexpression of GLS1 as a poor prognostic marker, while demonstrating that the inhibition of GLS1 with bis-2-(5-phenylacetamido-1, 3, 4-thiadiazol-2-yl) ethyl sulphide (BPTES), was associated with a reduced growth rate of tumours, highlighting the significance of glutaminolysis on the propagation of tumour cells [91]. An increased GLS1 mRNA level has also been seen in breast, gastric and oesophageal cancers. Zhu et al. found the use of a GLS1 inhibitor, CB-839, to significantly suppress cell proliferation and invasion of oesophageal squamous cell cancer [92]. Elevated GLS1 levels has also has been supported by studies on human glioblastomas and breast cancer, highlighting the central role glutaminolysis plays in tumorigenesis.

### 4.2. The Role of Glutaminase 2

Less is known about the hepatic-originating GLS2 and tumour cells. Although it is widely accepted, as outlined above, that an increased expression of GLS1 is pro-oncogenic, the role and direction of GLS2 is not as clear. GLS2 has been shown to be silenced in certain cancer cell lines, such as liver and colon cancers [93]. In glioblastoma cells, the overexpression of GLS2 actually slowed proliferation, altered decarboxylation and carboxylation of aKG in the TCA cycle and was associated with altered levels of nucleotides when metabolomics were studied [94].

### 4.3. Centromere Protein A as a Critical Transcription Factor

Centromere protein A (CENPA) has emerged as a critical transcription factor in the regulation of glutamine metabolism in endometrial cancer (EC) cells. Li et al. found CENPA to directly regulate the expression of the glutamine transporter SLC38A1, and subsequently enhance glutamine uptake, glutaminolysis and tumour progression. Conversely, the silencing of CENPA diminished glutamine metabolism and tumour progression [95]. *c-Myc* protein is an important oncogene that regulates gene expression. It has been associated with the ‘glutamine addiction’ of cancer cells. In cells with overexpressed *c-Myc*, glutamine deprivation accelerated fibroblast death, although this did not happen in controls [96]. This c-Myc-related glutamine addiction phenomenon was supported in osteogenic sarcoma cells [97], human B lymphoma cells [98] and small cell lung cancer cells [99]. *C-Myc* has also been found to control glutamine consumption in SF188 cells and also in breast cancer cells [100,101]. Furthermore, *c-Myc* has been found to contribute to glutamine-related biosynthetic processes. It has been shown to promote the synthesis of proline from glutamine [102], as well as aspartate and alanine by inducing expression of aspartate transaminase, GOT1, and alanine transaminase GPT2 [103,104]. In addition, cancers such as oesophageal squamous cell carcinoma frequently exhibit mTORC1 hyperactivation by Cyclin D1–CDK4/6, further integrating glutamine metabolism with cell growth and autophagy suppression [105]. Under hypoxia or in the context of mitochondrial dysfunction, tumour cells exploit glutamine-derived α-KG for citrate and lipid biosynthesis, underscoring the adaptability and importance of glutaminolysis in oncology.

### 4.4. Activating Transcription Factor 4 and Its Role in Glutaminolysis and Oncology

Activating transcription factor 4 (ATF4) is an important regulator of metabolic and oxidative stress. It is activated and elevated in response to cellular stress, and has a wide range of functions, including autophagy, the metabolism of amino acids and redox homeostasis [106]. It has been found to be upregulated in a range of tumours [107,108,109], and its expression increased in hypoxic environments [106]. Positive correlations have been found between the glutamine transporter SLC1A5 and the expression of ATF4 in oestrogen receptor-positive breast cancer [110]. This was supported by Ren and colleagues, who showed that the combined efforts of ATF4 and *N-Myc* can upregulate the transcription of the SLC1A5, and thus increase the dependency of neuroblastomas on glutamine [111]. In addition, the transcriptional co-activators YAP/TAZ directly regulate SLC1A5 transcription, and so YAP/TAZ depletion reduces glutamine metabolism (Kim et al., 2023 [112]). ATF4 is a transcriptional target of YAP/TAZ, thus relating mechanotransduction to metabolic adaptation in the form of glutaminolysis [113].

## 5. Glutaminolysis in Specific Systemic Fibrotic Diseases

### 5.1. Overview

Glutaminolysis and its role in profibrotic signalling is a known component of the pathophysiology of systemic fibrotic diseases. This metabolic pathway contributes to a variety of functions, including metabolic reprogramming and biosynthesis, and ultimately propagates these systemic fibrotic diseases.

### 5.2. Glutaminolysis in Cirrhosis

Myofibroblastic hepatic stem cells mirror the bioenergetic and biosynthetic requirements of proliferative cancer cells, including the use of glutaminolysis as an alternative energy source. Myofibroblastic transformation of hepatic fibroblasts occurs secondary to liver injury. Of course, a certain level is important for liver regeneration, but excessive levels cause fibrosis and can progress to liver cirrhosis. This progression of stem cells into myofibroblastic versions requires activation of both the hedgehog signalling pathway and Yes-associated protein 1 (YAP). Both of these proteins are upregulated in liver fibrosis and hepatic cellular carcinomas [114], with their relevant signalling contributing to metabolic reprogramming. Du et al. found myofibroblasts were highly dependent on glutamine, and that hedgehog-mediated YAP activation stimulated glutaminolysis and, subsequently, direct transdifferentiation from quiescent hepatic stem cells to a myofibroblastic version [115].

### 5.3. Glutaminolysis in Respiratory Fibrosis

Hamanaka et al. investigated the importance of glutamine metabolism in idiopathic pulmonary fibrosis and discovered that not only is its role in metabolic reprogramming important, but so too are its biosynthetic contributions to collagen protein synthesis. They found that culturing normal human lung fibroblasts void of glutamine inhibited TGF-β and induced the production of both collagen protein and α-SMA. Conversely, they also discovered that lung fibroblasts require the conversion of glutamate to proline in order for collagen protein production to occur [39]. By inhibiting ALDH18A1, a gene encoding P5CS that is responsible for the conversion of glutamate to proline, collagen protein production was significantly inhibited. Likewise, Choudhury et al. pharmacologically blocked SLC1A5 in fibrotic lung fibroblasts by using a small molecule inhibitor, V-9302. They found that this suppressed mTOR signalling and resulted in reduced oxidative phosphorylation and glycolysis as a result [116].

### 5.4. Glutaminolysis in Other Fibrotic Diseases

Intrauterine adhesion or, as it is histopathologically characterised, endometrial fibrosis is a common cause of infertility [117]. GLS1 and α-SMA expression has been found to be upregulated in fibrotic lesions when compared to healthy endometrium. Conversely, the inhibition of glutaminolysis suppressed α-SMA and COL-I expression, while alleviating endometrial fibrosis in a murine model [118]. Hewitson and Smith proposed the idea that renal fibroblasts are also metabolically reprogrammed, and the augmented glutamine metabolism contributes to renal myofibroblastic activation [36]. This was supported by Ou et al., who isolated glutaminolysis as a potential therapeutic target for kidney disease [71].

## 6. Glutaminolysis in the Eye

### 6.1. Glutaminolysis in the Cornea

The corneal endothelium is one of the most metabolically active tissues in the body. It maintains corneal clarity by controlling the hydration of the corneal stroma and actively pumping ions and fluid from the stroma into the anterior chamber across an osmotic gradient [119]. This function of ion and fluid transport necessitates a high metabolic activity. Work from Zhang et al. found that glutamine supports ATP for CE pump function, as well as contributing almost 50% of TCA intermediates. In addition, they found that the CE expresses glutamine metabolic enzymes, GLS1 and GLS2, while glutamine transporters such as SLC1A5 and SLC6A19 were also identified [120]. SLC4A11 is a membrane transporter that facilitates glutamine metabolism and suppresses mitochondrial superoxide [121]. It has been previously found to be highly expressed in the CE [122], while mutations of such have been associated with corneal endothelial dystrophies [123]. Of interest, it has also been found to be upregulated in some ‘glutamine-addicted’ cancers [124]. Elsewhere in the cornea, glutamine metabolism has been identified as a therapeutic target in dry-eye disease (DED) [125]. The pro-inflammatory and profibrotic properties associated with glutaminolysis were found to be of critical importance in the progression of dry-eye disease, and GLS1 expression has been found to be upregulated in the corneas of patients suffering from DED when compared to controls [126]. In addition, GLS1 has been found to be upregulated in the corneas of mice who experienced alkali burn injury and subsequent corneal neovascularisation. This was supported by the inhibition of GLS1 resulting in the suppression of corneal neovascularisation [127].

### 6.2. Glutaminolysis in the Lens

Glutathione is essential in maintaining lens transparency, and its precursors include cysteine, glycine and glutamine. ASCT2 (or SLC1A5) has been found to be expressed diffusely throughout the lens, including at the lens core [128]. In addition, research presented at the Association for Research in Vision and Ophthalmology Annual meeting 2023 found GLS to be upregulated in lens fibrosis, posing the hypothesis that lens fibrosis involves mitochondrial metabolic rewiring and glutaminolysis [129]. However, more robust evidence is needed before this is included in the literature.

### 6.3. Glutaminolysis in Ocular Diseases

Glutaminolysis has not been extensively researched in the eye. However, upregulated GLS1 and GLS2 have been found in the setting of subcapsular cataracts associated with hyperuricaemia. This suggests metabolic dysfunction may exist in the form of dysregulated glutaminolysis and, thus, glutamine-dependent redox imbalance that may influence cataract formation [130]. In age-related macular degeneration, serum glutamine was found to be increased, and the rate of glutaminolysis decreased [131], but larger metabolomic studies are needed to further investigate the metabolic profile of AMD patients. GLS1 expression has been found to be increased in the retinal pigment epithelial–choroidal complexes of rats with choroidal neovascularisation, a visually significant pathological change associated with various ocular diseases. This has been targeted with microRNA, miR-376b-3p, to successfully suppress glutaminolysis in human retinal microvascular endothelial cells. This poses evidence that upregulated glutaminolysis occurs in choroidal neovascularisation, and suggests a therapeutic method of targeting such [132]. Hypoxia and metabolic reprogramming have been referenced above, but interestingly, work from Singh et al. found that hyperoxia also leads to an accelerated rate of glutaminolysis in retinal Müller cells, with an upregulated GLS1 level [133].

## 7. The Important Fibrogenic Signalling Pathways Associated with Glutaminolysis

### 7.1. Transforming Growth Factor-Beta

Among the central mediators of fibrosis is TGF-β, particularly TGF-β1 [39], which orchestrates a profibrotic transcriptional programme through canonical and non-canonical signalling cascades. The inhibition of this factor can significantly limit fibrosis in a wide range of disease models [134]. The standard pathway involves TGF-β1, a profibrotic cytokine, binding to its serine/threonine kinase receptors (TGFBR1/2) [135], leading to phosphorylation of receptor-regulated SMADs (R-SMAD2 and SMAD3). These, in turn, complex with co-SMAD4 and translocate into the nucleus to regulate the expression of fibrogenic genes such as *COL1A1*, *FN1* and *ACTA2*, encoding collagen type I, fibronectin and α-SMA, respectively [136].

Emerging evidence signals that glutaminolysis is critical for the full transcriptional activity of TGF-β/SMAD signalling, and supports the role of glutaminolysis in fibrosis. The generation of α-KG via the glutaminase (GLS1)-mediated deamidation of glutamine is essential for the activity of α-KG–dependent dioxygenases [137], including Jumonji-domain histone demethylases [138] and Ten-Eleven Translocation (TET) enzymes, which regulate chromatin accessibility at fibrogenic loci [139]. Thus, α-KG acts as a cofactor enabling the TGF-β/SMAD-driven epigenetic priming of fibroblasts [140]. The inhibition of GLS1 has been shown to attenuate SMAD2/3-mediated transcription and reduce ECM gene expression [141], demonstrating that glutaminolysis is not just supportive but functionally integrated with TGF-β signalling.

### 7.2. MAPK Pathways

In parallel, the alternative TGF-β signalling pathways involving MAPKs (e.g., ERK, JNK, p38), PI3K/AKT, and Rho-like GTPases also contribute to fibroblast activation and metabolic reprogramming. These pathways can directly regulate glutaminase expression or activity, further linking glutamine metabolism with profibrotic signalling. For instance, TGF-β has been reported to upregulate GLS1 via mTORC1 activation in fibroblasts, reinforcing the metabolic flux through glutaminolysis and sustaining collagen production [142].

### 7.3. Energy Production Signalling

In addition, glutaminolysis contributes to redox homeostasis by supporting NADPH production through malic enzyme activity and the TCA cycle, reducing oxidative stress-induced apoptosis in activated fibroblasts [143]. This survival advantage facilitates chronic fibroblast persistence, a hallmark of progressive fibrosis [144]. The relevance of these mechanisms extends to multiple fibrotic diseases, including idiopathic pulmonary fibrosis (IPF) [145], liver fibrosis and renal fibrosis, where elevated GLS1 expression [90] and glutamine metabolism have been correlated with disease severity [146]. Targeting glutaminolysis through GLS1 inhibitors (e.g., CB-839) is currently under investigation as an antifibrotic strategy, with preclinical data demonstrating attenuation of fibrotic markers and tissue stiffness [147].

## 8. Targeting Glutaminolysis in Cancer and Fibrosis

### 8.1. Glutamine Antagonists

As outlined above, the metabolic mechanisms underlying glutamine dependence have been widely studied, as have their profibrotic effects (Table 1). Manipulating this knowledge is an important and exciting challenge in order to support patient care and explore disruptions to this metabolic pathway and its potential clinical applications. In murine cancer models, drugs that aim to augment glutamine transporters and their enzymes have exhibited significant therapeutic effects [148,149]. When glutamine antagonists inhibit both glutaminolysis and oxidative phosphorylation, they have been shown to reverse profibrotic changes. Preclinical studies of the glutamine antagonist 6-diazo-5-oxo-norleucine (DON) demonstrated potent antitumour activity across diverse cancer models, validating glutamine metabolism as a therapeutic vulnerability [150]. It has also been investigated in fibrosis, such as iatrogenic laryngotracheal stenosis, and was found to reduce collagen gene expression and proliferation rate [151]. However, DON’s lack of specificity proved to be a significant limitation due to its inhibition of a broad range of enzymes, and it elicited multiple toxicities, most notably in the gastrointestinal tract [152].

In contrast to DON’s broad inhibition, more targeted approaches have centred on GLS, the enzyme responsible for converting glutamine to glutamate. BPTES is a GLS1 inhibitor that results in the arrest of phosphorylation activation of GLS [153]. This has been shown to inhibit the proliferation of glioblastoma cells [154], while also successfully reducing the tumour weight of chondrosarcomas in mouse xenograft osteosarcoma models [155]. However, the clinical application of this inhibitor has proven challenging due to a low bioavailability and solubility [156]. Of note, a recent murine study found that targeting the deletion of GLS1 was sufficient to augment glutaminolysis and reverse fibrosis and cardiac dysfunction [55]. GLS2 inhibitors have not received the same attention as their GLS1 counterpart, although early trials have found them to have a potential role in anti-cancer therapies [157]. More research is needed in this area, as early results hint at its exciting potential as a therapeutic target.

### 8.2. Telaglenastat

CB-839 (telaglenastat) is a GLS1 inhibitor but with improved oral bioavailability and tumour suppression. It has been tested in a number of different oncology clinical trials, including in melanoma, triple-negative breast cancer and renal carcinoma. Clinical data suggest that the drug is well tolerated, with manageable side effects compared to DON [158], and shows enhanced antitumor activity when combined with other therapies. JHU083 is a DON prodrug that has been developed and proved to stem tumour cell growth in various tumours [159,160]. However, a pertinent challenge has been encountered in practical application, where the sole inhibition of glutaminase blockade leads to tumour cell resistance [161,162]. CB-839 was also found to successfully attenuate GLS1 in lung fibrosis [163], although further study is needed in this area.

### 8.3. Novel Therapies

Combination approaches have subsequently been considered, reflecting the recognition that tumours exhibit substantial metabolic flexibility, allowing them to compensate when one nutrient pathway is blocked. An example of this is the combination of CB-839 and metformin. Ren et al. found that this combination significantly inhibited osteosarcoma growth and metastasis by preventing compensation for metformin-induced electron transport chain inhibition through increased glutamine consumption, which subsequently produced other important molecules for cell proliferation [164]. This reiterates a recurring theme in cancer metabolism: that single-pathway inhibition rarely produces durable responses, while dual blockade can overwhelm the adaptive capacity of malignant cells. In addition, the dual inhibition of glycolysis and glutaminolysis has been found to be successful in targeting synovial fibroplasts in rheumatoid arthritis [165].

Transporters involved in glutaminolysis have been previously investigated as molecular targets in both cancer and fibrosis. SLC1A5 was found to be a promising therapeutic target and has been found to be inhibited by benzylserine and benzylcysteine [166]. SLC38A1, SLC38A2 and SLC6A14 are also emerging as glutamine-related transporters of interest [157,166,167].

Monoclonal antibodies have recently garnered attention in targeting glutamine transporters. Some of these have been designed to bind directly with SLC1A5 and, although they are in the early stages of development, have been shown to inhibit glutamine uptake and inhibit KRAS mutants [168,169]. Furthermore, glutaminolysis and an increased uptake of glutamine have been associated with certain cancers’ drug resistance to cisplatin, an oral chemotherapy agent.

**Table 1 ijms-27-00012-t001:** Glutaminolysis-targeting therapies previously investigated in the literature.

Therapeutic Mechanism	Drug/Molecular Target	References
Glutamine Antagonists	DON	Lemberg et al., 2018 [170]
		Lye et al., 2023 [171]
		Leone et al., 1979 [172]
	Acivicin	Kreuzer et al., 2015 [173]
		Olver et al., 1998 [174]
	Azaserine	Lyons et al., 1990 [175]
Glutamine Transporters	SLC1A5 (ASCT2)	Lye et al., 2023 [171]
		Kawakami et al., 2022 [176]
		Esslinger et al., 2005 [177]
		Chiu et al., 2017 [178]
	SLC38A1 (SNAT1)	Jalota et al., 2025; Liu et al. 2024 [179,180]
		Zavorka Thomas et al., 2021 [181]
	SLC38A2 (SNAT2)	Koe et al., 2025; Gauthier-Coles et al., 2022 [182,183]
	SLC6A14	Lu et al., 2022; Su et al., 2025; Coothankandaswamy et al., 2016 [184,185,186]
Glutaminolysis Enzyme Inhibitors	GLS1	Ramachandran et al., 2016 [187]
		Zimmermann et al., 2016 [188]
		Momcilovic et al., 2017 [189]
		Raczka and Reynolds, 2019 [190]
		Thompson et al., 2017 [191]
	GLS2	Lukey et al., 2019 [192]
	Glutaminase C	Katt et al., 2012 [153]

## 9. Fibrosis in Glaucoma

### 9.1. The Pathophysiology of Glaucoma and the Importance of Intraocular Pressure

Glaucoma, a group of progressive optic neuropathies, as outlined above, is a multifactorial group of diseases. The primary risk factor is age [71], which in itself is associated with an increased tissue stiffness within organs [193], and its relevance to glaucoma is outlined in the study by Liu et al. [194]. However, the most significant risk factor is an elevated IOP [195]. IOP is ultimately the pressure of fluid in the eye. It is the constant state of flux as aqueous humour is produced by the ciliary body and flows through one of two pathways against resistance (Figure 2). The majority of aqueous humour drains through the trabecular meshwork and into Schlemm’s Canal, where it reaches general circulation via episcleral veins [196]. The second pathway is known as the uveoscleral pathway, where AH flows through a suprachoroidal space and into the uveal venous circulation [197]. A study of the literature reveals alternative pathways that also contribute to AH fluid dynamics, including the recently hypothesised uveolymphatic pathway [198]. In an average eye, this pressure falls somewhere between 15 and 16 mmHg. The pathogenesis of this disease is intrinsically linked with fibrosis, involving fibrosis in the TM in the anterior chamber [199], Schlemm’s Canal and the LC at the optic nerve head [200] (Table 2).

### 9.2. Fibrosis in the Trabecular Meshwork

Pathological changes in the TM and Schlemm’s Canal (SC) endothelial cells include tissue stiffness [202] and ECM remodelling, which lead to increased resistance to aqueous outflow and contribute to increased IOP. Yang and colleagues recently investigated the development of fibrosis in glaucoma. Endothelial-to-mesenchymal transition is a process by which tissues may develop fibrosis, and is associated with various biomarkers, including α-SMA. These biomarkers were found to be upregulated in glaucomatous TM cells when compared to controls. In addition, glaucomatous samples were also found to have increased levels of FN-EDA fibronectin fibrils, proteins known to be elevated in pathological ECM production [203]. TGF-β plays a critical role in driving fibrogenesis in glaucoma. It is involved in numerous processes, so much so that treating TM and LC cells with TGF-β in vitro exhibits pathway activation profiles similar to those seen in glaucomatous tissues [204].

Integrin signalling plays a pivotal role in trabecular meshwork (TM) fibrosis through regulation of Rho family GTPases, which coordinate cytoskeletal remodelling, contractility, and extracellular matrix (ECM) deposition. In TM cells, αvβ3 integrin activates Rac1, promoting actin reorganisation into crosslinked actin networks (CLANs), a feature associated with glaucomatous TM [205]. Conversely, RhoA enhances actomyosin contractility and myofibroblast-like transformation largely through its downstream effector ROCK. In addition, α5β1 and αvβ3 integrins direct the assembly of fibronectin fibrils, which form the structural backbone of the ECM and facilitate the incorporation of additional matrix proteins [206]. In addition, studies in mice have found that αv-containing integrins may help promote the transformation of a cell into a myofibroblast by triggering the release of TGFβ1 from the surrounding ECM, while human TM cells have revealed that the contribution of αvβ3 integrin and connective tissue growth factor (CTGF) may also encourage the TGFB-induced myofibroblast [207].

### 9.3. Fibrosis in the Schlemm’s Canal

Profibrotic morphological and molecular changes have previously been found in glaucomatous Schlemm’s Canal endothelial cells. A higher level of α-SMA, collagen 1A1 (COL1A1) and TGF-B2, as well as an increased proliferation rate, was found when these endothelial cells when compared to normal human donors [208]. This supports both the ‘overgrowth’ and ‘increased stiffness’ phenomena seen in fibrosis [9]. In addition, the density of pores in the inner wall of these endothelial cells is thought to be lower than in controls, contributing to the increased resistance to aqueous flow and increased cytoskeletal stiffness [209]. This highlights the fibrotic contribution of glaucomatous SC cells to resistance in aqueous humour dynamics and increased intraocular pressure.

### 9.4. Fibrosis in the Lamina Cribrosa

The ECM of the LC in glaucomatous subjects has been found to be remodelled, contributing to the degeneration of retinal ganglion cell axons [210]. Hernandez and Ye found that glial hyperplasia contributes to an increased density of the area occupied by basement membranes in the LC of glaucomatous subjects. They found increased disorganisation of elastin fibres, and at an ultrastructural level, found abnormalities of the elastic fibre morphology, including increased fragmentation that alters the mechanical properties of the tissue. This was supported by the finding of decreased collagen fibre density in glaucomatous LCs [211]. Of note, the LC cells that uniquely contribute to the structure of the LC are phenotypically similar to myofibroblasts [212], and are integral to the cell signalling in wound healing and fibrosis [213]. Certain stresses, including hypoxia [214], substrate stiffness [215] and increased TGF-B [216], have been shown to upregulate the expression of genes involved in ECM synthesis. In addition, mitochondrial dysfunction contributes to the profibrotic gene expression in the lamina cribrosa [217]. This mitochondrial dysfunction is supported by the work of Kamel and colleagues, who investigated the alternative bioenergetic pathways preferred by mitochondria in glaucomatous subjects, and found evidence of decreased levels of oxidative phosphorylation and increased utilisation of alternative pathways [218]. When this is coupled with increased stiffness, this supports the myofibroblastic phenotype of LC cells, leading to a thinner, more fibrotic and stiffer LC [219]. These stiffer cell microenvironments have been associated with increased actin filament development and vinculin-focal adhesion formation, as well as changes in the cytoskeleton of glaucomatous LC cells, supporting evidence that stiffer cell microenvironments will successfully activate a myofibroblastic-like cell transformation in LC cells [194].

### 9.5. Glaucomatous Fibrosis and TGF-β

TGF-β plays a significant role in multiple fibrotic pathways in glaucoma, and although multiple cytokines and molecular signalling pathways are integral to glaucomatous fibrosis, TGF-β warrants a specific review. Tripathi and colleagues reviewed the level of TGF-β in the aqueous humour of patients with primary open-angle glaucoma and controls. They found the level of TGF-β to be significantly increased in patients with open-angle glaucoma [220]. These findings were also supported by Min et al. [221]. JD Pena and colleagues reviewed the levels of various isoforms of TGF-β in the optic nerve head, namely TGF-β1, TGF-β2 and TGF-β3, in foetal, normal adult and glaucomatous adult optic nerve heads [222]. They found no detectable immunoreactivity for TGF-β1 and TGF-β2. Foetal optic nerve head immunostaining revealed the presence of TGF-β1 in blood vessels, but not in glial cells, axons or other neural tissues. TGF-β2 was observed in neural tissues. No immunostaining occurred for TGF-β3. In glaucomatous optic nerve heads, TGF-β1 immunostaining was localised to small capillaries within the prelaminar and LC region, and was not found to have any presence in axons of the optic nerve head. Almost all cells were immunostained for TGF-β2 in the LC region. Immunostaining for TGF-β3 was negative. Zhavoronkov and colleagues also highlighted the importance of TGF-β in the activation of profibrotic pathways in the lamina cribrosa and trabecular meshwork, and found its role to be integral to the propagation of glaucoma [223].

**Table 2 ijms-27-00012-t002:** Evidence of fibrosis in various locations along the optic pathway in glaucoma. We have categorised the evidence in the table above corresponding to the histological, cellular and tissue levels for each location.

Ocular Tissue Type	References	Evidence
Trabecular Meshwork	Tamm et al., 2007; Yang et al., 2025; [203,206]	Increased accumulation of banded fibrillar elements derived from juxtacanalicular tissue. Increased deposition of FN-EDA fibronectin fibrils
Trabecular Meshwork	Callaghan et al., 2022 [204]	Treating cells with TGF-β induces the upregulation of profibrotic gene expression across a genome-wide transcriptome.
Trabecular Meshwork	Last et al., 2011; Keller et al., 2018 [202,224]	Mean elastic modulus of glaucomatous TM cells is significantly increased, leading to increased extracellular matrix stiffness
Schlemm’s Canal	Kelly et al., 2021. [208]	Increased actin stress fibres and density of F-actin cytoskeletal protein expression
Schlemm’s Canal	Kelly et al., 2021 [208]	Increased cell size and proliferation in glaucomatous SC endothelial cells
Schlemm’s Canal	Overby et al., 2014 [209]	Increased cytoskeletal stiffness of SC endothelial cells
Lamina Cribrosa	Hernandez et Ye, 1993; Liu et al., 2018 [194,211]	Increased collagen type IV mRNA. Increased actin filament development and vinculin-focal adhesion formation.
Lamina Cribrosa	Kirwan et al., 2009 [210]	ECM is remodelled and demonstrates increased profibrotic gene expression
Lamina Cribrosa	Zeimer et al., 1989 [219]	Stiffer, more fibrotic LC in glaucoma

## 10. Glutaminolysis in Glaucoma

### 10.1. Current Literature

Although the body of research regarding glutaminolysis in glaucoma is limited, its impact is significant. Recent research from our lab demonstrated evidence of metabolic reprogramming and mitochondrial dysfunction in human glaucomatous LC cells. An increased level of GLS2 was found, a marker for increased glutaminolysis, supporting the profibrotic metabolic signalling that occurs in glaucoma [218]. In addition, Pappenhagen et al. found that inducing stretch in optic nerve head astrocytes, such as that seen in glaucoma, revealed altered metabolic activity, although both control and stretched ONH astrocytes revealed a preference for glutamine as an energy source when compared to pyruvate and long-chain fatty acids [133]. Multiple studies have reviewed the metabolites in various biofluids within the eye and their relation to glaucoma. A review of metabolomic profiling of aqueous humour in glaucomatous patients from the Eye-D study found glutamine to be elevated [225]. This was supported by Lillo and colleagues [226] and also Breda and colleagues, who notably found the level of glutamine and α-KG, the final energy product of glutaminolysis, to be elevated in the aqueous humour of patients with glaucoma when compared to controls [227]. A plasma metabolite profile for primary open-angle glaucoma found glutamine to be nominally associated with a statistically significantly increased risk of glaucoma [228]. However, the evidence is inconsistent, as found by Wang et al., who systematically reviewed 18 metabolite articles and found divided opinions on the level of glutamine in various biofluids within the eye [229]. It is also worth noting that glutamine, along with taurine, has been found to be a significantly dominant amino acid in the tears of healthy individuals [230].

### 10.2. Glutaminolysis, Neurodegeneration and Glaucoma

Glaucoma has previously been termed a neurodegenerative disease by Gutpa et al. for the resulting loss of retinal ganglion cells and irreversible injury to the optic nerve [231]. We discuss this briefly in Section 3.5. Its associated optic nerve neurodegeneration has recently been found to share common neurodegenerative pathways with diseases such as Alzheimer’s Disease (AD) and Chronic Traumatic Encephalopathy (CTE) [232]. In addition, other investigations have found an increased metabolic vulnerability in the axons of glaucomatous optic nerves. This is attributed to the increased energetic demands, and energy disruption of glaucomatous axons, and the resulting reliance on alternative metabolic pathways [233]. Although the body of literature is sparse, glutaminolysis and glaucoma is an exciting area at present, which holds great potential for research, and necessitates further study to deepen our understanding of this profibrotic pathway in this disease intrinsically linked with fibrosis.

### 10.3. Potential Therapeutic Targets

Glutaminolysis has been targeted by both enzymatic inhibition and the blockade of transporters, as we have outlined above. However, there are not trials to our knowledge that involve glutaminolysis-targeted therapies for glaucoma at present. Elevated GLS1 and, to a lesser extent, GLS2 have shown therapeutic potential in terms of reducing glutaminolysis. They pose potential enzymatic targets for glaucoma and its metabolic reprogramming. In addition, combination therapies with glutaminolysis-related inhibitors and antifibrotic therapies may provide alternative therapeutic benefits and maximise synergistic effects [234], targeting both the metabolic reprogramming associated with glaucoma in the setting of glutaminolysis and its resulting fibrosis. Pretreatment of trabecular meshwork cells with nicotinamide riboside has been shown to have a protective effect against oxidative stress and fibrosis. It upregulated JAK2/Stat2 pathways but inhibited MAPK pathway expression, as well as exhibiting a lower rate of apoptosis and generation of ROS [235]. Kasetti and colleagues investigated ATF4 in glaucomatous TM cells and found its levels to be increased and its upregulated expression to lead to glaucomatous neurodegeneration [236]. An alternative study supported the role of ATF4 in glaucoma, as its deletion from retinal ganglion cells was found to promote axon survival and rescue glaucomatous neurodegeneration [237]. Given its role in glutamine metabolism as outlined earlier in the text, ATF4 may pose a synergistic target in augmenting glutaminolysis in glaucoma and altering glaucomatous neurodegeneration.

### 10.4. Future Directions

This encourages further study into the enzymes involved in glutamine metabolism and may present novel future targets. Further study into the glaucoma polygenic risk score has identified pyruvate as a ‘resilience biomarker’, and dietary supplementation was shown to have a protective effect against both intraocular pressure and optic nerve damage [238]. Although there is a lack of glaucoma-specific glutaminolysis studies, this review has explored the different aspects and pathways of glutaminolysis in fibrosis and its relevant association with glaucoma. In conclusion, glutaminolysis and its profibrotic signalling pathways pose an exciting new area in the study of glaucoma. More study is needed in tailoring the body of knowledge surrounding glutaminolysis to glaucoma in order to target this pathway, augment its associated fibrosis and explore new therapeutic avenues for patients.

## Figures and Tables

**Figure 1 ijms-27-00012-f001:**
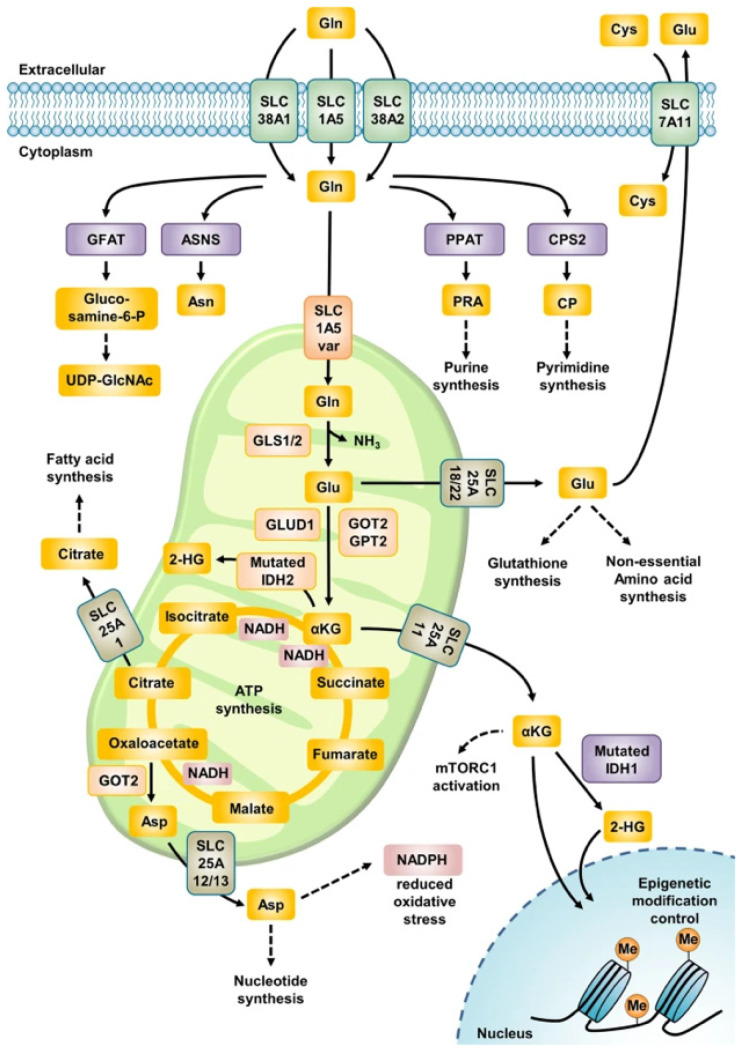
The metabolic pathway of glutaminolysis, showing glutamine uptake, transport into the mitochondria by SLC1A5 and integration into the TCA cycle [74].

**Figure 2 ijms-27-00012-f002:**
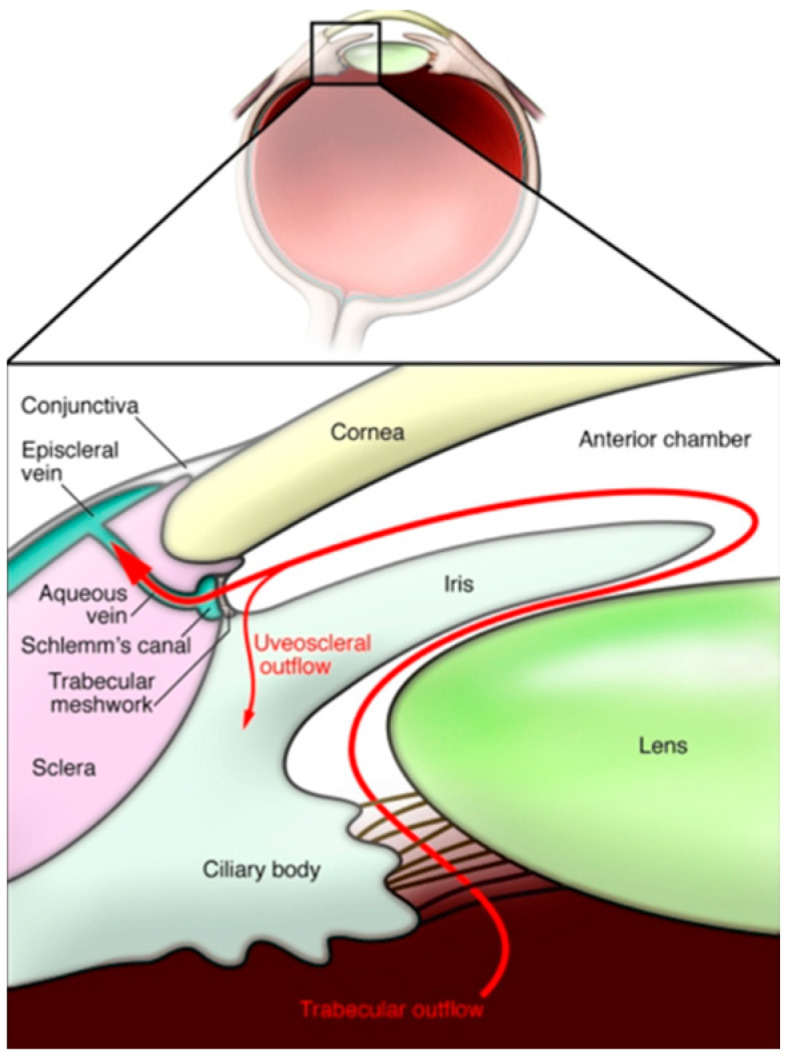
The structure and direction of flow of the trabecular and uveoscleral aqueous humour drainage pathway [201].

## Data Availability

No new data were created or analyzed in this study.

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
