# Peer review of "Targeting the Glutaminolysis Pathway in Glaucoma-Associated Fibrosis"

_ijms, 2025, doi:10.3390/ijms27010012_

Round 1

Reviewer 1 Report

Comments and Suggestions for Authors

Comments to the Authors

The present work was well-designed, and the manuscript was well-organized. This manuscript addresses an interesting and timely topic. I'd like to suggest a minor revision.

Below, I've included one major and several minor issues in the paper that need to be addressed.

Comments:

To explain fibrosis and metabolic reprogramming in fibrosis, it is necessary to describe the roles of cells (e.g. monocytes, T cells) and cytokines in this process.

Minor points:

1.  It is necessary to introduce abbreviations in the text. 

Abstract

Extracellular matrix (ECM)

Introduction

optic nerve head (ONH)

2. References

Check the citation of aeviation (EGCG) in the text.ll references in the paper, e.g. ref. 20, 122, 124.

Author Response

Many thanks for taking the time to review our paper. Your comments were much appreciated, and we have made the changes you suggested, as outlined below.

Comment 1: To explain fibrosis and metabolic reprogramming in fibrosis, it is necessary to describe the roles of cells (e.g. monocytes, T cells) and cytokines in this process.

Response 1: Many thanks for highlighting this oversight on our behalf. We agree with your comment and have included an additional section, 1.3, to discuss this point in the text. We have included the text below [p2, line 53-72], with references found in the manuscript:

1.3 Various Cell Types Associated with Fibrosis.

Multiple cell types are integral to the propagation of fibrosis, and almost all immune cell types are involved in some way [14]. Along with myofibroblasts mentioned above, macrophages, and their haematological precursor, monocytes, play an important regulatory role in fibrosis [15].  The diversity and plasticity associated with macrophages [16] is at odds with the stiff, over-proliferation associated with fibrosis.  However, their production of an inflammatory response, with the release of inflammatory chemokines such as Tumour Necrosing Factor-α, (TNF-α), and interleukin-1, -6 or -12 is an important host reaction to a hostile stimulus. When this reaction is not resolved, it can lead to chronic inflammation and fibrosis. Alternative types of macrophages can contribute to pathological scarring, propagating fibrosis [17].  In addition, it has recently been found that uncontrolled pro-inflammatory macrophages can directly contribute to the worsening of fibrosis through macrophage-myofibroblast transition (MMT) [18]. The role of T-cells in fibrosis is diverse, and different subgroups contribute to both the development and resolution of fibrosis. Their role is often tissue- and disease-specific.  Th1 cells produce pro-inflammatory cytokines that may attenuate fibrosis, but may play either a pro-fibrotic [19] or anti-fibrotic role, such as that seen in cardiac fibrosis [20]. Th2 cells can directly support fibrosis by stimulating fibroblasts in order to allow production of collagen, with IL-4 and IL-13 [21]. Th1 and Th2  T cells have been associated with scar-associated fibrosis [20]. Other types such as regulatory T cells may limit fibrosis by promoting tissue repair [22] .  

Comment 2: It is necessary to introduce abbreviations in the text. Abstract -  Extracellular matrix (ECM), Introduction- optic nerve head (ONH)

Response 2:  We have corrected the text to include these abbreviations, [pg 1 line 12, pg 1 line 38)

Comment 3: Check the citation of aeviation (EGCG) in the text.ll references in the paper, e.g. ref. 20, 122, 124.

Response 3: We have reviewed and corrected the references.

Reviewer 2 Report

Comments and Suggestions for Authors

The present submission, entitled “Targeting the Glutaminolysis Pathway in Glaucoma-Associated Fibrosis,” reviews glaucoma-associated fibrosis with a focus on glutaminolysis. At present, there is very limited literature reviewing this theme. The manuscript is structured into nine sections: Introduction; Metabolic Reprogramming in Fibrosis and Cancer; Glutaminolysis; Glutaminolysis in Oncology; Glutaminolysis in Specific Systemic Fibrotic Diseases; Important Fibrogenic Signalling Pathways Associated with Glutaminolysis; Targeting Glutaminolysis in Cancer and Fibrosis; Fibrosis in Glaucoma; and Glutaminolysis in Glaucoma.

As reflected by the headings, there is extensive discussion on cancer. The authors state that they review the evidence for fibrosis and metabolic reprogramming in oncology and systemic fibrotic diseases, that reveals a predilection for glutaminolysis.” (lines 17 and 18). They also note that research on glutaminolysis is mainly limited to various types of cancer (line 82). However, there is already ample literature on metabolic reprogramming and glutaminolysis in ocular cells (see below). Therefore, why did the authors choose to focus on cancer cells rather than ocular cells when the review’s core theme is an ocular disease? A more comprehensive literature search in ocular research is recommended and should be discussed in the review. 

A brief rationale should be included in the Introduction or Section 2 to outline the similarities between oncology and fibrosis in terms of pathogenesis with a citation. This would provide justification for extrapolating research findings from oncology to fibrosis.

Some references on glutaminolysis in ocular cells are as follows:

  • Glutaminolysis is Essential for Energy Production and Ion Transport in Human Corneal Endothelium. EBioMedicine. 2017 Feb;16:292-301. doi: 10.1016/j.ebiom.2017.01.004. Epub 2017 Jan 13.
  • MicroRNA-376b-3p Suppresses Choroidal Neovascularization by Regulating Glutaminolysis in Endothelial Cells. Invest Ophthalmol Vis Sci. 2023 Jan 3;64(1):22. doi: 10.1167/iovs.64.1.22.
  • Reprogramming metabolism by targeting sirtuin 6 attenuates retinal degeneration.J Clin Invest. 2016 Dec 1;126(12):4659-4673. doi: 10.1172/JCI86905. Epub 2016 Nov 14.

Section 7, “Targeting Glutaminolysis in Cancer and Fibrosis” contains very limited coverage on fibrosis. Apart from ocular fibrosis, what about systemic fibrosis? 

The final section, “Glutaminolysis in Glaucoma,” does not provide comprehensive coverage. While section 9.1/9.2 elaborates on clinical findings, discussion of relevant cell types or animal models involved in glaucoma such as Müller cells and retinal ganglion cells, should also be included to further support the role of glutaminolysis in glaucoma. For example,

  • Effect of adenosine and adenosine receptor antagonist on Muller cell potassium channel in Rat chronic ocular hypertension models. Sci Rep. 2015 Jun 11;5:11294. doi: 10.1038/srep11294.
  • Effect of SCH442416 on glutamate uptake in retinal Müller cells at increased hydrostatic pressure. Mol Med Rep. 2015 Sep;12(3):3993-3998. doi: 10.3892/mmr.2015.3882. Epub 2015 Jun 3.
  • 5-HT1A Receptor Agonist Promotes Retinal Ganglion Cell Function by Inhibiting OFF-Type Presynaptic Glutamatergic Activity in a Chronic Glaucoma Model. Front Cell Neurosci. 2019 May 3;13:167. doi: 10.3389/fncel.2019.00167. eCollection 2019.
  • Stretch stress propels glutamine dependency and glycolysis in optic nerve head astrocytes. Front Neurosci. 2022 Aug 5;16:957034. doi: 10.3389/fnins.2022.957034.

Figures and tables should be cited appropriately in the text. The table headings should be revised to better categorize the content. For example, the column heading “Authors” should be replaced with “References” and positioned as the last column. Table 1 does not clearly convey key messages, include relevant cell type. In addition, since the first column of Table 2 lists ocular structures (e.g., trabecular meshwork), the column heading should be revised to “Ocular Tissue Type.”

Author Response

Response To Peer Review 2:

Many thanks for taking the time to review our paper. You made astute and relevant comments, and we feel your contribution has improved the quality of our paper. We really appreciate this. Following our review of your comments, we have addressed such below.

Comment 1: As reflected by the headings, there is extensive discussion on cancer. The authors state that they review the evidence for fibrosis and metabolic reprogramming in oncology and systemic fibrotic diseases, that reveals a predilection for glutaminolysis.” (lines 17 and 18). They also note that research on glutaminolysis is mainly limited to various types of cancer (line 82).  However, there is already ample literature on metabolic reprogramming and glutaminolysis in ocular cells (see below). Therefore, why did the authors choose to focus on cancer cells rather than ocular cells when the review’s core theme is an ocular disease? A more comprehensive literature search in ocular research is recommended and should be discussed in the review. 

Response 1: Many thanks for this comment. We thought it accurately highlighted a deficiency in our paper and subsequently included an additional section, 6. Glutaminolysis in the Eye (page 9, line 425 – page 10, line 472).  References are found in the manuscript.

6.1 Glutaminolysis in the Cornea

The corneal endothelium is  one of the most metabolically active tissues in the body. It maintains corneal clarity by controlling hydration of the corneal stroma and actively pumping ions and fluid from the stroma into the anterior chamber across an osmotic gradient [118]. This function of ion and fluid transport necessitates a high metabolic activity. Work from Zhang et al, found that glutamine supports ATP for CE pump function, as well as contributing almost 50% of TCA intermediates. In addition, they found that the CE expresses glutamine metabolic enzymes, GLS1 and GLS2, while glutamine transporters such as SLC1A5 and SLC6A19 were also identified [119].  SLC4A11 is a membrane transporter that facilitates glutamine metabolism and suppresses mitochondrial superoxide [120] . It has been previously found to be highly expressed in the CE[121], while mutations of such have been associated with corneal endothelial dystrophies [122]. Of interest, it has also been found to be upregulated in some ‘glutamine-addicted’ cancers [123].  Elsewhere in the cornea, glutamine metabolism has been identified as a therapeutic target in dry-eye disease (DED)[124] . The pro-inflammatory and pro-fibrotic properties associated with glutaminolysis were found to be of critical importance in the progression of dry eye disease and GLS1 expression has been found to be upregulated in the corneas of patients suffering with DED when compared to controls [125]. In addition, GLS1 has been found to be upregulated in corneas of mice who experienced alkali burn injury, and subsequent corneal neovascularistion. This was supported by the inhibition of GLS1 resulting in the suppression of corneal neovascularisation [126]. 

6.2 Glutaminolysis in the Lens

Glutathione is essential in maintaining lens transparency, and its precursors include cysteine, glycine and glutamine. ASCT2 (or SLC1A5) has been found to be expressed diffusely throughout the lens, including at the lens core [127].  In addition, research presented at the Association for Research in Vision and Ophthalmology Annual meeting 2023 found GLS to be upregulated in lens fibrosis, posing the hypothesis that lens fibrosis involves mitochondrial metabolic rewiring and  glutaminolysis [128]. However, more robust evidence is needed before this is included in the literature.

6.3 Glutaminolysis in  Ocular Diseases

Glutaminolysis has not been extensively researched in the eye. However, upregulated GLS1 and GLS2 have been found in the setting of subcapsular cataract associated with hyperuricaemia. This suggests metabolic dysfunction may exist in the form of dysregulated glutaminolysis and thus, glutamine-dependent redox imbalance that may influence cataract formation [129]. In age-related macular degeneration, serum glutamine was found to be increased, and the rate of glutaminolysis  decreased [130], but larger metabolomic studies are needed to further investigate the metabolic profile of AMD patients.  GLS1 expression has been found to be increased in retinal pigment epithelial-choroidal complexes of rats with choroidal neovascularisation,  a visually significant pathological change associated with various ocular diseases. This has been targeted with microRNA, miR-376b-3p, to successfully suppress glutaminolysis in human retinal microvascular endothelial cells. This poses evidence that upregulated glutaminolysis occurs in in choroidal neovascularisation, and suggests a therapeutic method of targeting such [131]. Hypoxia and metabolic reprogramming has been referenced above, but interestingly, work from Singh et al found that hyperoxia also lead to an accelerated rate of glutaminolysis in retinal Müller cells, with an upregulated GLS1 level [132].

Comment 2: A brief rationale should be included in the Introduction or Section 2 to outline the similarities between oncology and fibrosis in terms of pathogenesis with a citation. This would provide justification for extrapolating research findings from oncology to fibrosis.

Response 2: Many thanks for pointing this out. We did not make our point clear, and your comment has helped to clarify this. Following a review, we decided it was necessary to include a section in the introduction outlining the similarities between the pathogenesis in fibrosis and cancer. This is found in section 1.5, Similarities between Fibrosis and Cancer, (page 2-3, lines 85-101). References available in the manuscript.

1.5 Similarities between Fibrosis and Cancer

Fibrosis and cancer share multiple similarities, not only in their accumulation of and changes within the ECM, but also in the presence of genetic alterations and metabolic reprogramming [26]. Fibroblast proliferation and sustained differentiation are pathophysiological mechanisms common to both fibrosis and cancer. This in turn leads to ECM modifications and increased ECM stiffness [27], [28]. With regards genetic alterations, both fibrosis and cancer have been found to have increased transforming growth factor-beta (TGF-β), production and cellular senescence [29]. In addition, epithelial-mesenchymal transition (EMT), is a biological process where epithelial cells transform both phenotypically and functionally to mesenchymal-like cells [30], and this is a process common to both entities. Pro-fibrotic and pro-inflammatory cytokines support the process of EMT in fibrosis [31] . Likewise in cancer, EMT supports the proliferation, seeding and development of metastasis [32]. The presence of these shared properties is supported by pharmaceutical exploitation of such, such as the targeting of the AXL tyrosine kinase receptor. This has been shown to reduce fibrosis [33], while also exhibited promising anti-tumour effects [34]. Metabolic reprogramming is intrinsic to the proliferation of cancer cells, and is also common to fibrosis. This will be discussed in detail below.

Comment 3: Section 7, “Targeting Glutaminolysis in Cancer and Fibrosis” contains very limited coverage on fibrosis. Apart from ocular fibrosis, what about systemic fibrosis? 

Response 3: Thank you for your comment. Glutaminolysis in fibrosis is discussed in detail in section 5 of the original paper, including sub-headings of cirrhosis, respiratory and ‘other fibrotic diseases’, such as endometriosis and kidney disease. We found that, in the literature, there has been more research into the targeting of glutaminolysis in the context of cancer than in fibrosis. However, this research is often not specific to cancer, and pertains more to augmentation of the glutaminolysis pathway in general. This is supported by similar drugs being investigated in both fibrosis and cancer, and working through the same augmention of the glutaminolysis pathway (eg CB-839). We reviewed the literature again, for evidence of therapeutic targets in the context of fibrosis and have included examples we thought relevant. References pertaining to fibrosis include reference 36, 39, 71, 113-117, and are available in the manuscript. We have added the following text to the paper:

Section 8.1, (page 11, line 526-528):  It has also been investigated in fibrosis, such as iatrogenic laryngotracheal stenosis, and found to reduce collagen gene expression and proliferation rate [150]. 

Section 8.2:, (page 12, line 552-553): CB-839 was also found to successfully attenuate GLS1 in lung fibrosis [188], although further study is needed in this area.

Section 8.3, (page 13, line 573-577): Transporters involved in glutaminolysis have been previously investigated as molecular targets in both cancer and fibrosis. SLC1A5 was found to be a promising therapeutic target, and has been found to be inhibited by benzylserine and benzylcysteine [156]. SLC38A1, SLC38A2 and SLC6A14 are also emerging as glutamine-related transporters of interest [156], [189] [190].

Comment 4: The final section, “Glutaminolysis in Glaucoma,” does not provide comprehensive coverage. While section 9.1/9.2 elaborates on clinical findings, discussion of relevant cell types or animal models involved in glaucoma such as Müller cells and retinal ganglion cells, should also be included to further support the role of glutaminolysis in glaucoma.

Response 4: Many thanks for this comment. You went to great effort to source papers for us, and we really appreciate this. Glutamate, and excitotoxicity relating to glaucoma has been readily discussed in the literature (reference 81-89), especially in retinal ganglion cell death by apoptosis . It is intrinsically linked to the neurodegeneration associated with glaucoma. However, our paper wished to focus on the metabolic reprogramming associated with glaucoma, specifically glutaminolysis. Therefore, although we really appreciate this comment, we do not think the inclusion of those references in the section on glutaminolysis in glaucoma are directly relevant as the majority deal with glutamate, glutamate and excito-toxicity and neurodegeneration. We reviewed our sections again, and added a reference on glutamine metabolism and optic nerve head astrocyte stretch in section 10.1, (page 16, line 738 – 742).

In addition, Pappenhagen et al found that inducing stretch to optic nerve head astrocytes, such as that seen in glaucoma, revealed altered metabolic activity, although both control and stretched ONH astrocytes revealed a preference for glutamine as an energy source when compared to pyruvate and long chain fatty acids [132].

Given the extensive body of work in the literature, pertaining to glutamate and neurodegeneration, and following your comments, we have included a section on neurodegeneration as below.

3.5 Neurodegeneration

Glutamate, of which glutamine is the main precursor,  is the primary excitatory neurotransmitter in the brain, and has many roles in the function of the central nervous system [81]. It is inextricably linked to glutaminolysis, as glutamine is deaminated in mitochondria by glutaminase to produce glutamate [82]. Glutamatergic dysregulation has long been recognised as a significant factor in numerous disorders, including psychiatric, neurodevelopmental, and neurodegenerative disease [83], [84], [85]. An excess of glutamate can flood neuronal glutamate receptors, and ultimately lead to neuronal oxidative stress [86].

Its toxicity, and resulting neurodegeneration is known to exist within the eye since the work of Lucas and Newhouse in 1957, who discovered its toxic effect on the inner layers of the retina [87]. The exact mechanism of this so-called excitotoxicity is unclear. Previously, varying levels of glutamate were found to cause both apoptotic and necrotic injury[88]. Wu et al recently revisited its role on excitotoxicity, and found it to be multi-factorial, involving calcium imbalance, increased oxidative stress, resulting increase of misfolded proteins and the induction of apoptosis, as well as the disruption of cellular homeostasis [89].

Although its role has been previously investigated in the setting of glaucoma, no definitive results, or therapeutic targets have been discovered.  However, this paper will focus on the role of glutaminolysis in metabolic reprogramming, and not its relation to the neuro-excitotoxic properties of glutamate, for which there is existing extensive literature.

Comment 5: Figures and tables should be cited appropriately in the text. The table headings should be revised to better categorize the content. For example, the column heading “Authors” should be replaced with “References” and positioned as the last column. Table 1 does not clearly convey key messages, include relevant cell type. In addition, since the first column of Table 2 lists ocular structures (e.g., trabecular meshwork), the column heading should be revised to “Ocular Tissue Type.”

Response 5: We have made changes as you have suggested. We have included the title Drug/Molecular Target in table 1 to help  clearly convey our key message that these are the drugs utilised or molecules targeted during the various glutaminolysis-targeting therapies. In addition we have replaced authors with references, and renamed the first column ocular tissue type in table 2.

Round 2

Reviewer 2 Report

Comments and Suggestions for Authors

The authors have satisfactorily addressed all comments. Thank you!